Subject Areas:
biomechanics, evolution

Keywords:
macroevolution, biomechanics, multi-body dynamics, finite element analysis, rodent mastication

Author for correspondence:
Karl T. Bates
e-mail: k.t.bates@liverpool.ac.uk

# Evolutionary biomechanics: hard tissues and soft evidence?

Sarah Broyde[1], Matthew Dempsey[1], Linjie Wang[4], Philip G. Cox[2,3], Michael Fagan[4] and Karl T. Bates[1]

[1]Department of Musculoskeletal Biology, Institute of Aging and Chronic Disease, University of Liverpool, The William Henry Duncan Building, 6 West Derby Street, Liverpool L7 8TX, UK
[2]Department of Archaeology, and [3]Hull York Medical School, University of York, PalaeoHub, Wentworth Way, Heslington, York YO10 5DD, UK
[4]Department of Engineering, University of Hull, Hull HU6 7RX, UK

LW, 0000-0003-2661-9533; PGC, 0000-0001-9782-2358; KTB, 0000-0002-0048-141X

Biomechanical modelling is a powerful tool for quantifying the evolution of functional performance in extinct animals to understand key anatomical innovations and selective pressures driving major evolutionary radiations. However, the fossil record is composed predominantly of hard parts, forcing palaeontologists to reconstruct soft tissue properties in such models. Rarely are these reconstruction approaches validated on extant animals, despite soft tissue properties being highly determinant of functional performance. The extent to which soft tissue reconstructions and biomechanical models accurately predict quantitative or even qualitative patterns in macroevolutionary studies is therefore unknown. Here, we modelled the masticatory system in extant rodents to objectively test the ability of current muscle reconstruction methods to correctly identify quantitative and qualitative differences between macroevolutionary morphotypes. Baseline models generated using measured soft tissue properties yielded differences in muscle proportions, bite force, and bone stress expected between extant sciuromorph, myomorph, and hystricomorph rodents. However, predictions from models generated using reconstruction methods typically used in fossil studies varied widely from high levels of quantitative accuracy to a failure to correctly capture even relative differences between macroevolutionary morphotypes. Our novel experiment emphasizes that correctly reconstructing even qualitative differences between taxa in a macroevolutionary radiation is challenging using current methods. Future studies of fossil taxa should incorporate systematic assessments of reconstruction error into their hypothesis testing and, moreover, seek to expand primary datasets on muscle properties in extant taxa to better inform soft tissue reconstructions in macroevolutionary studies.

## 1. Introduction

Changes in functional morphology have underpinned some of the most significant evolutionary transitions in the history of life. Colonization of the land by the earliest tetrapods [1], mammalian origins and diversification [2–5], the evolution of locomotion in dinosaurs and birds [6–23], and functional and ecological shifts in human ancestors [24–31] represent extensively studied examples. The last two decades has seen widespread adoption of sophisticated mathematical-computational approaches to study functional morphology in extinct animals and the biomechanics of evolutionary transitions documented in the fossil record. These approaches realize a number of benefits relative to more traditional comparative approaches [32,33], particularly the ability to deliver absolute measures of functional performance in fossil animals (e.g. energy costs, maximal performance), thereby allowing quantitative tests of how anatomical innovations enabled major behavioural niche adaptions over geological time.

Mathematical-biomechanical approaches yield quantitative predictions of animal performance by combining general models of Newtonian physics and solid mechanics with mathematical descriptions of tissue behaviour and physiology. In doing so, they incorporate all the major causative anatomical and physiological factors that underpin mechanical function, and in living animals, these approaches have been shown to deliver accurate predictions of metabolic energy costs in walking (e.g. [25]), maximal locomotor (e.g. [8,13,21]) and bite performance (e.g. [34,35]) among other parameters. However, one challenging aspect in their use on extinct animals is that they require precise specification of numerical values for soft tissue parameters that are rarely, or never, preserved in fossils. Studies of extinct animals have subsequently employed a diverse range of approaches to estimate absolute values for soft tissue parameters in fossil organisms, ranging from estimated mean values for living taxa (e.g. [12,13,19–21,34]), scaling values from analogous extant animals (e.g. [12,13,24–26,28–31]), extrapolating values from estimated muscle attachment areas [e.g. 10,27,36,37], and computer-aided design approaches to reconstruct the size of soft tissues directly in the fossil themselves (e.g. [5,14–16,22,23,34,35,38]). However, it remains uncertain what the likely error magnitudes are for such soft tissue reconstructions. It is therefore unclear whether or not the uncertainty surrounding soft tissue parameters is yielding such significant errors that biomechanical studies lack the resolution required to accurately reconstruct the functional consequence of anatomical change and test hypotheses about macroevolutionary radiations observed in the fossil record.

In this study, we take the most direct and comprehensive approach to date to assess how inaccuracy in soft tissue reconstruction currently impact upon our ability to identify quantitative and qualitative differences between extinct taxa, and therefore our ability to recognize adaptive trends and evolutionary changes in the fossil record. To do this, we first use real (measured) soft tissue data to carry out multiple types of biomechanical modelling on extant taxa that are known to exhibit quantitative and qualitative functional differences. Subsequently, we repeat this multi-modal biomechanical analysis by substituting real (measured) soft tissues properties with values derived from reconstructive methods typically used on fossil animals. Comparing the functional predictions generated using 'real' versus reconstructed soft tissue data not only allows us to examine inaccuracy quantitatively, but perhaps more fundamentally allows us to examine if known qualitative differences between extant taxa are preserved by current soft tissue reconstruction methods. This ability to reliably identify qualitative differences between extinct taxa is fundamental to evolutionary studies that seek to identify adaptations or trends across fossil lineages and major evolutionary transitions in the history of life [1–69]. Prior to this study, this fundamental premise, underpinning an entire field of research [1–69], has not been extensively tested.

## 2. Material and methods

### (a) Case study: evolutionary biomechanics of the rodent masticatory system

The Rodentia is the largest order of extant mammals, comprising over 2500 living species. Despite this diversity, almost all rodents can be assigned to one of three groups based on the morphology of their masticatory musculature, specifically the masseteric complex. These three morphotypes are all thought to be derivations of the ancestral morphology (present in a single living species, the mountain beaver), and are referred to as the 'sciuromorph' (squirrel-like), 'myomorph' (mouse-like), and 'hystricomorph' (porcupine-like) conditions [70–72]. Each of these derived morphotypes represents an extension of the masseter on to the rostrum: in sciuromorph species, the lateral masseter originates from an expanded zygomatic plate; in hystricomorphs, the zygomatico-mandibularis extends through the orbit and an enlarged infraorbital foramen; and myomorphs show a combination of both the sciuromorphous and hystricomorphous conditions [70–72]. Each muscle arrangement has evolved at least twice independently within the rodents, and previous analyses have indicated that each conveys different functional capabilities i.e. sciuromorphy enables efficient gnawing at the incisors, hystricomorphy leads to efficient molar chewing, and myomorphy provides greatest efficiency at both feeding modes [71,72]. Thus, the rodents are an ideal case study for testing the accuracy with which muscle anatomy can be estimated from skeletal morphology, and the impact of such estimations on inferences of function.

### (b) Quantitative soft tissue reconstructions

Our soft tissue reconstructions focus on two critical parameters that govern muscle force generation and subsequently play a highly determinate role in bite force magnitudes and the magnitude and distribution of stress/strain in the skull: muscle volume and fibre length (FL). Under static maximal biting conditions typically analysed in fossil taxa, muscle force is calculated according to

$$\text{Muscle force} = \text{physiological cross-sectional area (PCSA)}$$
$$\times \text{ maximum isometric stress.}$$
$$(2.1)$$

With muscle volume and FL determining the physiological cross-sectional area (PCSA) in parallel-fibred muscles according to

$$\text{Muscle PCSA} = \frac{\text{muscle volume}}{\text{muscle FL}}. \quad (2.2)$$

And in pennate muscles according to

$$\text{PCSA} = \frac{\text{muscle volume}}{\text{muscle FL}} \times \text{COS (pennation angle).} \quad (2.3)$$

Here, we developed a protocol for muscle volume sculpture (figure 1a) based on methods used in previous fossil studies (e.g. [34,35,56,67]). This protocol was formalized in an instruction sheet (see electronic supplementary material), which outlined the specific modelling approach to be used and anatomical diagrams on which to base the three-dimensional muscle sculptures around three-dimensional bone models. Previous application of similar methods to the same fossil specimens by independent research teams has produced highly disparate muscle volumes [38]. We therefore conducted the first analysis of inter-investigator variability in muscle volume sculpture, with three of the authors independently generating muscle volumes in all three rodent models following only the instruction sheet. A brief discussion of investigator expertise and experience is provided in the electronic supplementary material.

Different approaches to muscle FL estimation has also led to highly disparate functional predictions in extinct animals [38,69]. Here, we used three approaches used in a recent study [38], which cover different assumptions about the nature of muscle architecture in the extinct group under analysis. First, we generated FLs for each muscle under the assumption that all

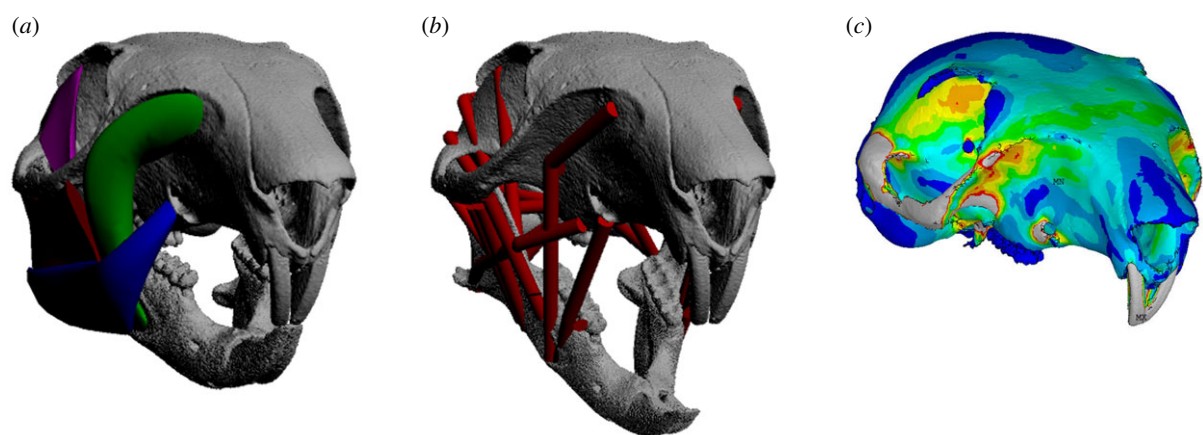

**Figure 1.** Quantitative soft tissue reconstruction and biomechanical modelling of rodent masticatory morphotypes. (*a*) Muscle volumes are reconstructed using three-dimensional sculpture techniques, as commonly applied in fossils, with values combined with different estimates of fibre length to provide input values for bio-mechanical models. Incisor bite forces were predicted across 27 'fossil' model iterations of (*b*) MDA models for comparison to values predicted using real (measured) muscle data. (*c*) Predicted muscle forces from all model iterations were used to load FE models to compare stresses predicted in fossil models to those from models with real (measured) muscle properties. (Online version in colour.)

muscles were non-pennate (i.e. parallel fibred), and that FLs were equal to muscle length. In this scenario (hereafter referred to as iteration A), the PCSAs of all muscles are calculated according to equation (2.2). Second, we generated an iteration of models which differed only in their specification of the medial pterygoid muscle. This muscle consistently shows a pennate architecture in rodents [70], with pennation angles of 20–25 degrees in the three taxa studied here (electronic supplementary material, tables S1–S3). Our second iteration of the models (iteration B) therefore represented the medial pterygoid muscle with a pennation angle of 25 degrees in all three taxa, with calculated PCSA for this muscle according to equation (2.3). The average ratio of measured FL to muscle length across the three taxa was used to calculate the FLs for the medial pterygoids in this iteration. Finally, in a third iteration (iteration C), all muscles were modelled as pennate, with a pennation angle of 25 degrees, the maximum value measured in these three rodents. The average ratio of measured FL to muscle length in each muscle across the three taxa was used to calculate the FLs for all muscles and subsequently PCSA (using equation (2.3)) for this iteration. These three FL and PCSA iterations were applied to the three muscle volume sculptures generated independently by the three investigators, yielding nine fossil models per taxon to be evaluated relative to the model using real (measured) muscle values in multi-body dynamics (MDA) and finite element (FE) models.

### (c) Multi-body dynamics analysis

We used the forwards dynamic package GaitSym (version 2013) to construct MDA models (figure 1*b*) and simulate maximal, symmetrical incisor bite forces in all three rodent models following the approach of [35,38] (see also the additional description in electronic supplementary material). We generated 10 MDA models for each taxon. For each taxon, we generated an 'extant' model, where muscle FLs and PCSAs were derived directly from specimens being modelled [70]. The remaining nine models consisted of three per investigator, in which each investigator's muscle volumes were used to generate three models according to the three fibre architecture iterations (A, B, and C) explained above. All soft tissue input values for the 27 fossil iterations are tabulated in electronic supplementary material, tables S7–9.

### (d) Finite element analysis

We re-analysed the existing FE models [71,72] of incisor biting in our three rodent taxa in ANSYS Mechanical APDL 2019 R1 using

the newly generated muscle force values from our MDA models (figure 1*c*). As far as possible, models remained as described in [71,72], with only minor modifications made in the conversion to ANSYS (see the electronic supplementary material). To compare the stresses predicted by the different model iterations, we uniformly divided each cranium into 10 sections anteroposteriorly (electronic supplementary material, figure S3). The mean von Mises stress of all elements in each section was extracted and calculated for every loading scenario's simulation. FE models, and the extant iterations of our MDA models, are available to download from https://datadryad.org/stash/share/6uhYkXexzlJGK6e5zMSWuUuBwnMbrF-ruFbGMa87iwo and https://doi.org/10.17638/datacat.liverpool.ac.uk/1184.

## 3. Results

### (a) Muscle volume reconstruction

The total (summed) masticatory muscle mass reconstructed by investigator 1 yielded errors of 14.5%, 9.7%, and 3.1% for the guinea pig, rat, and squirrel (figure 2 and electronic supplementary material, tables S4–S6). Investigator 2 produced lower errors of 1.8%, 3%, and –2.8% for the guinea pig, rat, and squirrel, while investigator 3 produced greater errors of 57.8%, 15.3%, and 93.8% (figure 2 and electronic supplementary material, tables S4–S6). Error magnitudes for individual muscles varied more widely, from less than 1% up to 552% (figure 2 and electronic supplementary material, tables S4–S6). Visual inspection suggests no common pattern among muscles in terms of error magnitudes, although on the whole there was a greater tendency to overestimate rather than underestimate muscle volume (figure 2 and electronic supplementary material, tables S4–S6). Regression analysis provides no support for size effects (e.g. systematically larger errors in bigger or smaller muscles) in error magnitudes (electronic supplementary material, figure S4).

The three investigators also vary considerably in relative accuracy of the reconstructed total muscle volume and the relative volumes of individual homologous muscles across the three species. Measurements indicate that guinea pigs have the highest summed masticatory muscle volume, followed by the squirrel and then the rat. Investigators 1 & 2 recovered this relative pattern correctly, but the reconstructions by

Proc. R. Soc. B 288: 20202809

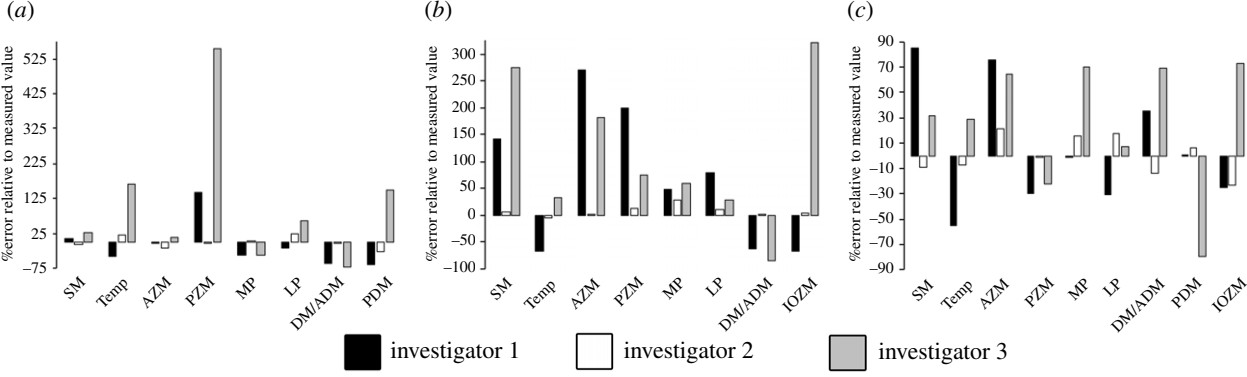

**Figure 2.** Error magnitudes in the sculptured muscle volume reconstructions by investigators 1, 2, and 3 for the (*a*) squirrel, (*b*) guinea pig, and (*c*) rat. Abbreviations: SM, superficial masseter; Temp, temporalis; AZM, anterior zygomatico-mandibularis; PZM, posterior zygomatico-mandibularis; MP, medial pterygoid; LP, lateral pterygoid; DM/ADM, deep masseter/anterior deep masseter; PDM, posterior deep masseter; Infraorbital zygomatico-mandibularis.

investigator 3 produced qualitative error with the squirrel being reconstructed with greater overall masticatory muscle volume than the guinea pig. In terms of the relative sizes of individual muscles, investigator 1 produced 36% correct relative placements, versus 84% and 52% in the reconstructions of investigators 2 & 3.

## (b) Muscle fibre length and physiological cross-sectional area

Muscle architecture iteration A overestimated muscle FL in all muscles in this analysis (figure 3 and electronic supplementary material, tables S11–S13). That is, muscle length always exceeded measured FLs in the masticatory muscles of all three taxa. Overestimation ranged from +55% to +205% in the squirrel, +29% to +292% in the guinea pig, and +20% to +203% in the rat (figure 3 and electronic supplementary material, tables S11–S13). By using the average muscle length to FL ratio to derive FL, muscle architecture iteration C yielded much lower errors in predicted FLs, with errors ranging from −27.3% to +40%, −6.6% to +86.4%, and −42.84% to +17.5% in the squirrel, guinea pig, and rat (figure 3 and electronic supplementary material, tables S11–S13).

Because PCSA is a function of muscle volume and FL, and muscle volume varied considerably and non-systematically across the investigators (figure 2), this parameter shows a complex pattern across the fossil model iterations (figure 3 and electronic supplementary material, tables S11–S13). However, on the whole, muscle architecture iteration A tended to underestimate PCSA in all models (all species, all investigators) even where investigators had overestimated muscle volume (figures 2 and 3) due to the relatively large errors resulting from the assumption that FL was equal to muscle length (figure 3). Maximum underestimations of PCSA were quite similar across species (−81.7% to −96%) and all occurred in models of investigator 3. Where overestimation of PCSA did occur in iteration A, investigator 3 again yielded the highest errors in all three species, with magnitudes of +283.6%, +94.1%, and +39.13% in the squirrel, guinea pig, and rat (figure 3 and electronic supplementary material, tables S11–S13). The range of PCSA error magnitudes in models using muscle architecture iteration C was greater (figure 3 and electronic supplementary material, tables S11–S13), despite the fact that this iteration matched real (measured) FLs more closely than iteration A (figure 3

and electronic supplementary material, tables S11–S13). The range in error magnitudes varied considerably across the three species, ranging from −80.5% to +714%, −92.3% to +240.5%, and −65.1 to +80.3% in the squirrel, guinea pig, and rat (figure 3 and electronic supplementary material, tables S11–S13).

Investigator 1 correctly ordered individual taxa in terms of relative PCSA seven out of 24 (29%) times in their muscle architecture iteration A, and eight out of 24 (33.3%) times in iteration C. Despite relatively high quantitative errors, investigator 3 correctly ordered individual taxa in terms of relative PCSA 18 out of 24 (63%) times in both muscle architecture iterations A and C. In line with their relatively lower absolute errors in PCSA, investigator 2 correctly ordered individual taxa in terms of relative PCSA 18 out of 24 (75%) times in both muscle architecture iterations A and C.

## (c) Bite forces

Our initial MDA models, using measured muscle properties yielded maximal static incisor bite forces of 47.9 N, 56.8 N, and 70.2 N for the guinea pig, rat, and squirrel models (figure 4 and electronic supplementary material, table S14). The three model iterations of investigator 1 yielded quantitative errors in incisors bite force ranging between −65.9% and +16.9% of the extant models. All model iterations from investigator 2 underestimated bite force, by between −63% to −6.7%, while the models reconstructed by investigator 3 ranged from −52.2% to +30.6% of the values from the extant models (figure 4). Within each investigator, the lowest bite forces and largest absolute errors were recovered in iteration A, where the overestimation of FLs yielded underestimates of PCSA and subsequently maximum isometric muscle force (figure 4 and electronic supplementary material, tables S11–S13). Reconstructing the medial pterygoid with more representative pennate architecture and shorter FLs led to only very small improvements (1–5%) in absolute accuracy (figure 4 and electronic supplementary material, table S14). Applying this approach to FL estimation and PCSA calculation to all muscles (iteration C) led to underestimation in bite force in investigator 2 being reduced to between −6.7 and −17.6%, and overall error in investigator 1 to −18.6% to +9.8% across the three taxa (figure 4 and electronic supplementary material, table S14). However, in investigator 3, iteration C reversed the −35 to −62% underestimated error seen in iterations A and B to slightly lower magnitudes of

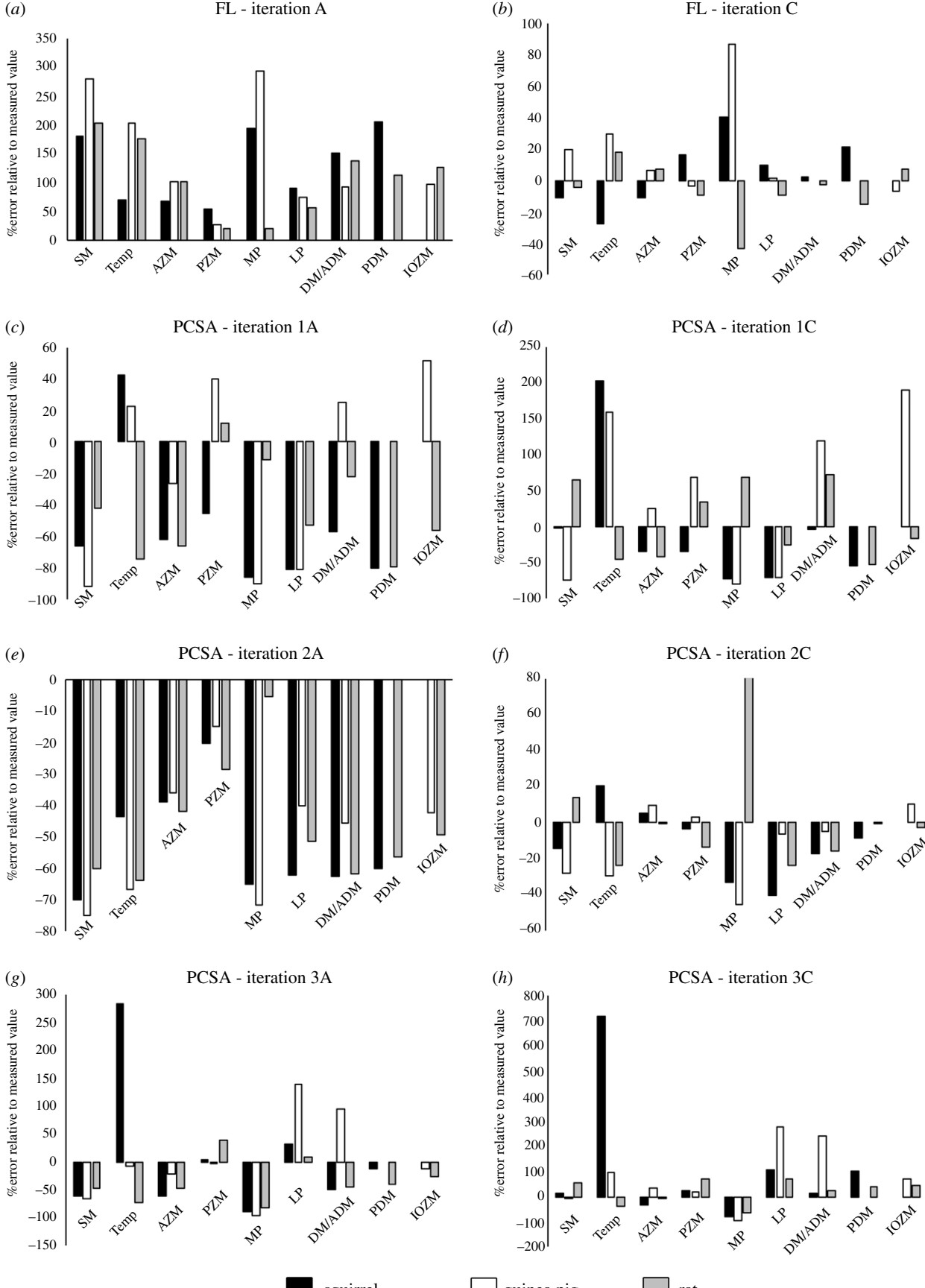

**Figure 3.** Error magnitudes in reconstructed (a,b) muscle fibre lengths and (c–h) PCSAs in the three species.

overestimated error (+13 to +30.6%; figure 4 and electronic supplementary material, table S14).

The three investigators also vary considerably in the accuracy with which their models correctly predicted the relative bite forces of the three species. None of the model iterations

generated by investigator 1 placed all three taxa in the correct order in terms of relative bite force. Investigator 1's models did consistently predict higher bite forces in the rat compared to the guinea pig, but only iteration C correctly predicted higher forces in the squirrel compared to the guinea pig.

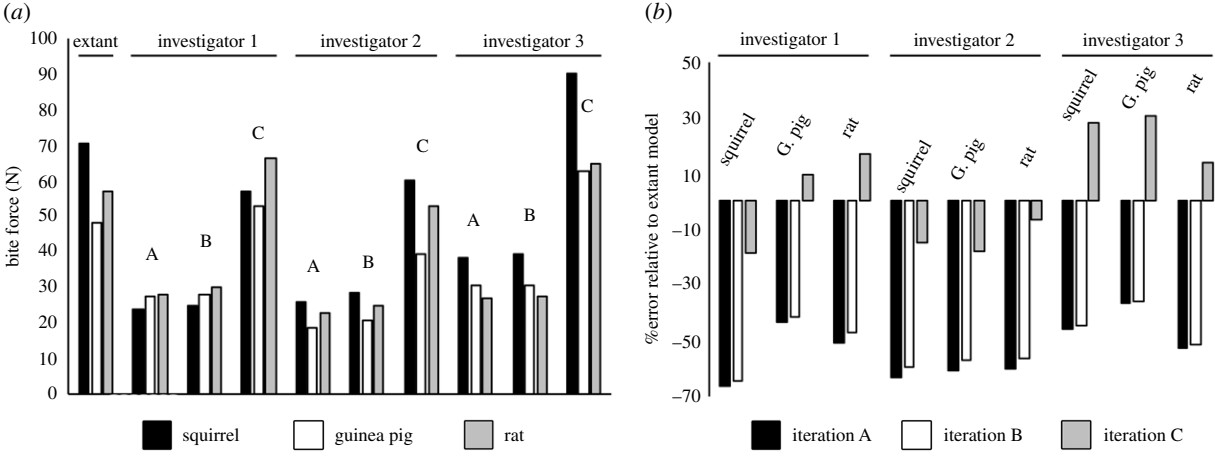

**Figure 4.** Comparison of (a) absolute bite forces and (b) percentage error magnitudes in bite forces across the 'extant' and 'fossil' MDA models. (a) 'Extant' model iterations predict the highest incisor bite forces in the squirrel, followed by the rat and then guinea pig. This qualitative pattern across the morphotypes is recovered in all model iterations by investigator 2, by iteration C investigator 3, but in none of the iterations by investigator 1. (b) Quantitative error varied considerably, with most iterations tending to underestimate bite force.

Iterations A and B by investigator 3 correctly identified the squirrel as generating the highest bite force of the three taxa, but incorrectly predicted relatively higher bite forces in the guinea pig compared to the rat. Iteration C by investigator 3 and all three iterations (A–C) by investigator 2 correctly predicted relative bite forces across the three species.

## (d) Stress and strain in finite element models

FE models loaded using outputs from the 'extant' MDA models indicate that the rat experiences the highest stresses, followed by the squirrel and then the guinea pig along the entire skull length (figure 5a–d). The most striking pattern among fossil model iterations is the variation in stress magnitudes. With the exception of small regions of the rat and guinea pig models in iteration C of investigator 2 (figure 5b,d,e), all fossil models produced by investigators 1 and 2 underestimate stress relative to the extant models (figure 5a,b). Error is higher in the models of investigator 1, where stress magnitudes are less than one-third of that seen in extant models in some regions of the skull (figure 5a,d,f). The models of investigator 3 showed a more complex pattern of error, with all model C iterations overestimating stress magnitudes throughout the skull, while iterations A and B vary in the nature and magnitude of error across the three rodent taxa (figure 5c). For example, iterations A and B of the guinea pig model slightly underestimate stress in most regions, but overestimate stress in between 30 and 45% skull length (figure 5c).

Despite extremely high variation in stress magnitudes, the qualitative pattern or distribution of stress across the skull seen in the extant models is mostly preserved in the fossil model iterations (figure 5 and electronic supplementary material, figure S5) with relatively subtle deviations. A notable exception to this is the absence of the sharp increase in stress, or stress peak, between 20 and 50% skull length in all three fossil iterations of the squirrel model of investigator 1, which changes the stress distribution in the zygomatic arch relative to the extant model and the models of the guinea pig and rat (figure 5). This error in the squirrel models of investigator 1, along with general underestimation of stress therein, means that the relative stress patterns recovered in the squirrel and guinea pig are qualitatively reversed (figure 5a,d,f). The models of investigator 3 mostly preserve qualitative

differences between the morphotypes, but iteration C exaggerates the quantitative differences, while iterations A and B underestimate them (figure 5c).

## 4. Discussion and conclusion

Soft tissue reconstructions and biomechanical models provide quantitative measures of functional performance in extinct taxa and thereby offer a unique insight into major behavioural or niche adaptions over geological time and selective pressures driving major evolutionary radiations [1–68]. In this study, we have taken a novel approach to evaluating the absolute and relative accuracy of soft tissue and biomechanical reconstructions of extinct animals, and the ability of current methods to accurately capture a functional macroevolutionary radiation (figures 2–5). The rodent masticatory system has evolved three distinct morphotypes (sciuromorph, hystricomorph, and myomorph) with osteological, myological, and functional characteristics that lead to disparate specializations in food processing in each morphotype. The rat, the representative of the myomorph condition, has a temporalis muscle 1.6× larger than the squirrel (sciuromorph) and 1.7× than the guinea pig (hystricomorph) [71]. Despite this significant difference in size, only one of the three investigators sculpted the rat with the largest temporalis muscle and ordered the three morphotypes successfully in relative temporalis size (figure 2). The medial and lateral pterygoids were also reconstructed disproportionately in relative terms by all three investigators: two of the three investigators correctly reconstructed the guinea pig with the largest medial pterygoid, but incorrectly reconstructed the squirrel as having the smallest volume for this muscle (figure 2). The other investigator incorrectly reconstructed the squirrel with the largest medial pterygoid and rat with the smallest (figure 2). None of the investigators correctly reconstructed the squirrel with the largest lateral pterygoid volume (figure 2). However, despite often large magnitudes of quantitative error (figure 2), the qualitative proportions of a number of muscles (e.g. posterior deep masseter, posterior and infraorbital zygomatico-mandibularis) were correctly reconstructed by two and sometimes all three investigators. Overall, the investigators averaged 70.3%, 12.3%, and 94.57% quantitative error in volume at the

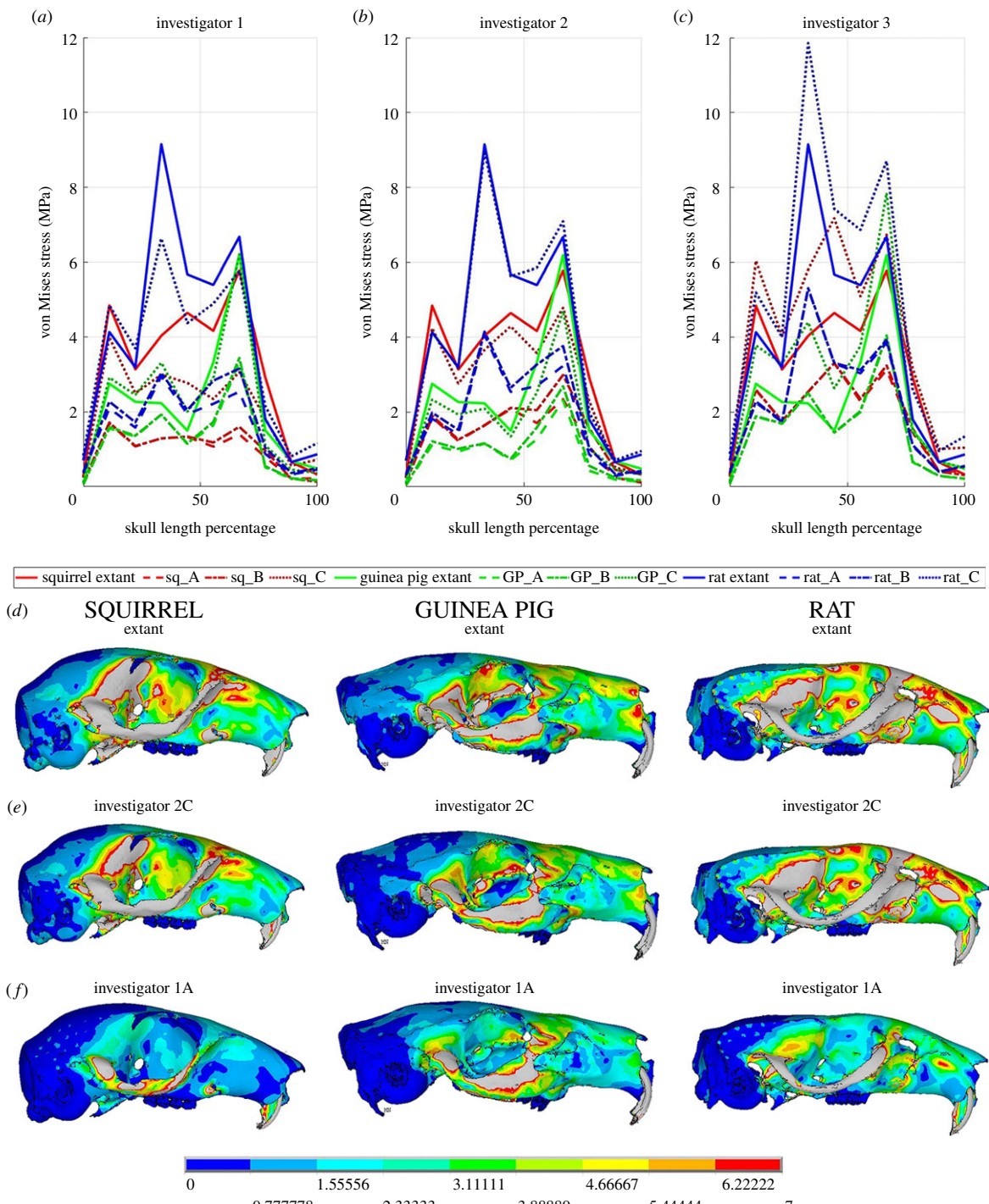

**Figure 5.** Stress magnitudes and distributions (represented by von Mises stress) in the FE models across the 30 model iterations. Stress magnitudes along the length of skull in the extant models are compared to those of (*a*) investigator 1, (*b*) investigator 2, and (*c*) investigator 3 and demonstrate significant quantitative and some qualitative error. Some reconstructions, such as (*b,e*) iteration C those by investigator 2, show a close quantitative match to (*d*) the extant models, while some reconstructions, such as (*f*) iteration A by investigator 1 contain both quantitative and qualitative error in relative stress magnitudes and distribution across the morphotypes. (Online version in colour.)

individual muscle level (figure 2), providing clear evidence that studies using volume sculpture approaches to assess the evolution of muscle proportions and performance should incorporate an assessment of the error in their hypothesis testing.

Bite force, and the mechanical efficiency of biting, are crucial adaptive functional distinctions between the three rodent morphotypes [71,72]. Our extant MDA models with real muscle properties predict the highest incisor bite forces in the squirrel, followed by the rat and then guinea pig (figure 4), which is consistent with previous studies [71,72].

Here, we show, for the first time, that accuracy with which such a qualitative macroevolutionary pattern is recovered by palaeontological methods varies across investigators and across different model iterations according to the reconstruction of muscle architecture (figure 4). The impact of subjectivity, largely related to the sculpture of muscle volumes (figure 2), is manifested in the highly disparate relative accuracy in bite forces across the investigators: investigator 1 did not capture the true macroevolutionary pattern in any iteration, while investigator 2 correctly recovered the expected pattern across morphotypes in all cases (figure 2). This

difference reflects the considerably lower levels of qualitative and quantitative error in muscle volumes sculpted by investigator 2 (figure 2). However, the pattern of relative error in bite force seen in investigator 3 demonstrates that even recovering qualitative differences between taxa is not simply a matter of accurately reconstructing muscle size. Muscle force is proportional to PCSA (equation (2.1)), which is a function of muscle volume and fibre architecture (equations (2.2) and (2.3)). Model iterations A and B of investigator 3, in which muscles are reconstructed with FLs equivalent to muscle length, led to incorrect relative bite forces and failure to capture the true functional macroevolution pattern that has evolved across rodent morphotypes (figure 2). However, the use of average ratios of muscle FL to overall length to calculate FL led to investigator 3's muscle volumes correctly recovering the true macroevolutionary pattern across rodent morphotypes (figure 2). This emphasizes the complex interaction between the estimation of muscle size, architecture, and force-generating capabilities, and highlights that simple sensitivity tests in which muscle size or force is scaled uniformly up or down may be insufficient in macroevolutionary studies.

These issues regarding both quantitative and qualitative error in masticatory muscle anatomy and bite force translate directly into analyses of absolute and relative stress in FE models (figure 5). To our knowledge, this is the first study to explicitly examine the likely magnitudes of error in FE models capturing a macroevolution radiation resulting from disparate reconstructions of muscle force-generating properties. As with muscle volumes (figure 2) and bite forces (figure 4), our data provides clear evidence that current approaches to soft tissue reconstruction can not only recover the correct qualitative or relative differences between taxa, but also generate stress magnitudes and distributions that are quantitatively consistent with models loaded using real (measured) muscle data (figure 5b,d,e). While this is encouraging, the large errors noted in muscle volume, architecture, and bite force predictions (figures 2–4) inherently mean that many of the fossil model iterations yield highly inaccurate stress magnitudes and, in some instances, produce magnitudes and distributions that are qualitatively dissimilar to the extant models and thus do not correctly capture the true qualitative macroevolutionary pattern (figure 5a,d,f). Cox et al. [71] noted that stress patterns along the zygomatic arch are different between the three rodent morphotypes, which our extant models capture here (figure 5a–d). The magnitude of the stress differences in this region of the skulls varies across model iterations, particularly those of investigator 3 where relative differences between rodents are exaggerated and underestimated by different iterations (figure 5c). Underestimation of stress in the zygomatic arch in the models of investigator 1 means that the relative stress magnitudes between the squirrel and guinea pig models are incorrectly represented in this key region (figure 5a,d,f). Cox et al. [71] also note that the rat shows a pattern of elevated stress around the origin of the temporalis muscle compared to the guinea pig and squirrel models, which is causatively associated with this taxon's larger temporalis muscle (figure 2). The extent to which this pattern is recovered in the fossil models presented here varies according to the accuracy of temporalis muscle reconstruction. As noted above, only one of the investigators correctly reconstructed the relative size of the temporalis muscle across the three rodent morphotypes (figure 2).

To put our study and its conclusions into context, we surveyed 68 published studies that used quantitative soft tissue reconstruction alone or in combination with biomechanical models to examine evolutionary changes in functional morphology in fossil taxa [1–31,34–69]. Our goal was not to provide exhaustive coverage of all relevant papers, but to sample enough studies to provide coverage of most major taxonomic groups, body regions (limbs, skulls, necks etc.), and methodological approaches. Our subjective assessment of this literature leads us to suggest that only around 35% of studies have used methods of numerical soft tissue reconstruction that have been validated for precision and accuracy in extant animals, and only around 32% of studies have used any kind of sensitivity analysis in their assessments of the force-generating capacity of muscles in extinct animals. In the latter aspect (sensitivity analysis), this figure of 32% can be considered optimistic as we chose to be maximally inclusive and include studies that our present results (figures 2–5) would suggest are insufficient in terms of sensitivity testing. For example, a number of assessments of bite mechanics in extinct animals provided minimum and maximum estimates of bite force by either selecting extreme low and high values for maximum isometric stress [43,44] or by adding a model iteration in which a correction factor was applied to increase muscle force [45] across all muscles. As our results demonstrate, uniform error in the reconstruction of individual muscles, even within one taxon, should not be expected (figures 2 and 3), and the magnitude of non-uniform error across muscles results in unpredictable and differential consequences in functional predictions for bite force and stress magnitudes (figures 4 and 5). Breaking these studies down in body regions and biomechanical approaches reveals a clear signal in the tendency to quantitatively validate and recognize soft tissue error in biomechanical predictions. Studies of limbs more frequently applied at least some of their reconstructions approaches to extant animals (approx. 90%) and carried out sensitivity analyses on their reconstructions of fossil taxa (approx. 55%), while studies of skulls have done so much less frequently (approx. 7% and approx. 21%, respectively). This same disparity is reflected in MDA (approx. 70% and approx. 45%) versus FEA (approx. 3% and approx. 17%) approaches because the majority of locomotor studies have used MDA, while FEA is most common in analyses of skulls.

The quantitative error will perhaps always remain unavoidable in evolutionary biomechanics, but an ability to identify qualitative similarities and differences across fossil lineages, and between extinct taxa and extant groups with known behaviours is fundamental to our understanding of palaeoecology and ecosystem dynamics, adaptive radiations and selective extinctions, and functional constraints on biological evolution [1–69]. Our novel analysis highlights that correctly reconstructing qualitative differences between taxa in a macroevolutionary radiation is challenging and that both false positive and negative results are possible using current approaches to quantitative soft tissue reconstruction. Our results provide quantitative evidence that studies of fossil taxa should incorporate a systematic assessment of reconstruction error into their experimental procedures and hypothesis testing, and provide a clear incentive for an expansion of primary datasets on muscle properties in extant taxa to better inform soft tissue reconstructions in macroevolutionary studies.

Data accessibility. All models are available from the Dryad Digital Repository: https://doi.org/10.5061/dryad.kd51c5b4c [73].

Authors' contributions. K.T.B. conceived the study. S.B., K.T.B., and P.G.C. designed the study. S.B., K.T.B., M.D., L.W., M.F., and P.G.C. collected the data and carried out the analyses. All authors contributed to the manuscript.

Competing interests. We declare we have no competing interests.

Funding. This work was funded by a NERC standard grant (NE/G001952/1) to P.G.C. and M.J.F., a BBSRC responsive mode grant to M.F. and K.T.B. (nos. BB/R016380/1; BB/R016917/1; BB/R017190/1) and a NERC doctoral dissertation grant (nos. NE/S00713X/1) to M.D.

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
