## [Peer Review File · Proceedings of the Royal Society B: Biological Sciences]

Review History

RSPB-2020-2809.R0 (Original submission)

Review form: Reviewer 1

Recommendation

Accept as is

Scientific importance: Is the manuscript an original and important contribution to its field?

Excellent

General interest: Is the paper of sufficient general interest?

Good

Quality of the paper: Is the overall quality of the paper suitable?

Good

Is the length of the paper justified?

Yes

Should the paper be seen by a specialist statistical reviewer?

No

Do you have any concerns about statistical analyses in this paper? If so, please specify them explicitly in your report.

No

It is a condition of publication that authors make their supporting data, code and materials available - either as supplementary material or hosted in an external repository. Please rate, if applicable, the supporting data on the following criteria.

Is it accessible?

Yes

Is it clear?

Yes

Is it adequate?

Yes

Do you have any ethical concerns with this paper?

No

Comments to the Author

Broyde and colleagues examined the effects of interobserver variability and muscle reconstruction method on the outcomes of comparative MDA and FEA of three rodent taxa. They found substantial variation arising from observer- and reconstruction method-specific outcomes. The results clearly demonstrate a hitherto unstudied and unquantified source of error in those simulation-based methods. The paper is well written, and I have no comments to add to the paper. I applaud the authors for completing an important study to further improve the applicability and replicability of MDA and FEA as applied in functional morphological and paleontological studies.

Review form: Reviewer 2

Recommendation

Reject – article is not of sufficient interest (we will consider a transfer to another journal)

Scientific importance: Is the manuscript an original and important contribution to its field?

Acceptable

General interest: Is the paper of sufficient general interest?

Acceptable

Quality of the paper: Is the overall quality of the paper suitable?

Acceptable

Is the length of the paper justified?

Yes

Should the paper be seen by a specialist statistical reviewer?

No

Do you have any concerns about statistical analyses in this paper? If so, please specify them explicitly in your report.

No

It is a condition of publication that authors make their supporting data, code and materials available - either as supplementary material or hosted in an external repository. Please rate, if applicable, the supporting data on the following criteria.

Is it accessible?

Yes

Is it clear?

Yes

Is it adequate?

Yes

Do you have any ethical concerns with this paper?

No

Comments to the Author

Broyde and co-authors present a new methodological approach to discuss the implications about soft tissue reconstruction. It's a good idea and well-written. They use 3 common mammals as case study to evaluate problems to reconstruct soft tissues and the implications to reconstruct these tissues in fossils.

The paper seems to be the consequence of a problem arised in a previous work (Bates and Falkingham, 2018 J Anat). I agree with them that soft tissue reconstruction needs further works to obtain proper and refined results, but the submitted approach requires additional work and to be re-thought prior to be published.

On this sense, I think that the manuscript present some flaw points and possibly to re-focus the paper if authors wants to publish it on a high impact journal as Proc Roy Soc B:

-Number of investigators: this is one of the most flaw points: all work is based only on 3 investigators who independently reconstructed soft tissues. Authors recognize that this is a major limitation of the work. The bias is so important and the results can not be considered as valid just using three diferent investigators with different experience on modelling and myology.

-Iterations: it's unclear the steps followed for the 3 iterations. I reccommend to add further data / figure / protocol sheet. Not sure if it's mostly based on muscle sculpture (and due to the low number of investigators previously mentioned, again, biased).

-Discussion and Conclusions: Regarding FEA implications, I'm not sure to fully agree with authors, as to perform FE analyses, it is usually required insertion areas, not using muscle volume reconstruction. This is particularly remarkable in 2D models but also in other cases, so authors needs to re-evaluate the citations used to then discuss properly what it has been done in the literature.

- Discussion and Conclusions: authors include some % data about the published papers that used different variables (i.e. sensitivity) but they don't provide information about how they calculated these percentages. Is it based only on the cited publications? If yes, how can you validate that these citations are representative of all published works?

Other issues:

Line 43:"in dinosaurs and birds": As this work is focused in mammals, I think that just few citations are enough.

Line 85: "not been extensively tested", please give details (i.e. citations) about tests performed to date.Might I also reccommend to down the tone of the sentence.

Review form: Reviewer 3

Recommendation

Accept with minor revision (please list in comments)

Scientific importance: Is the manuscript an original and important contribution to its field?

Excellent

General interest: Is the paper of sufficient general interest?

Good

Quality of the paper: Is the overall quality of the paper suitable?

Excellent

Is the length of the paper justified?

Yes

Should the paper be seen by a specialist statistical reviewer?

No

Do you have any concerns about statistical analyses in this paper? If so, please specify them explicitly in your report.

No

It is a condition of publication that authors make their supporting data, code and materials available - either as supplementary material or hosted in an external repository. Please rate, if applicable, the supporting data on the following criteria.

Is it accessible?

Yes

Is it clear?

Yes

Is it adequate?

Yes

Do you have any ethical concerns with this paper?

No

Comments to the Author

I think that this is an important and long-overdue paper. Certainly, it makes for essential - if uncomfortable - reading for those performing biomechanical modelling on extinct (or even poorly-known extant) taxa. That said, it is actually surprising how well the FEA analyses do at replicating relative stress patterns given the great uncertainty in muscle forces - this is also a very significant result (and something of a relief) in itself. The inclusion of suggestions for best practices going forwards, as drawn from quantitative results, is appreciated and maximise the utility of this paper for the field.

Fortunately, the paper is well-executed - I appreciate the author's efforts both not to over-complicate the experimental setup, and to include a varied but realistic range of investigator experience levels and data availability to replicate palaeontological analyses. I am hence happy to recommend it for publication, pending three very mild points, as listed below.

First: how accurately were lines of action estimated by each of the investigators? This is another potential source of substantial error in modelling loading regimes of extinct taxa and so, if it varied between investigators, it would be good to see it reported. I know there is less margin of error for this in rodents than in larger and taller skulls, and the consistency of fibre length results further suggest that your methodology minimised the chances of these errors, but it would still be good to verify this. Even if little-to-no variance was seen here for these reasons, it would still be worth noting that this will remain yet another source of significant source of error in other taxa (e.g. large sauropsids) beyond that seen in your small, well-bracketed mammals here.

Second, it is stated in the discussion that 0% of cranial fea studies have also applied their reconstruction techniques to extant taxa, and implied throughout that comprehensive looks at muscle properties have not been previously considered. This seems a little harsh given that, for example, Cost et al. (2020) recently combined validation work on extant lizards and parrots to ground estimates of muscle pennation angles and fibre length in their work on *Tyrannosaurus rex*. I know this was not to the same extent or with the same experimental setup or goals as you have performed here, but it at least is fair to note that some FEA work on the skulls of extinct taxa has attempted to bracket soft tissue parameters in a systematic way. (Reference: Cost, I.N. et al. (2020) Palatal biomechanics and its significance for cranial kinesis in *Tyrannosaurus rex*. *The Anatomical Record*, 303, 999-1017.

Finally, although the main text Figure 5 is currently fine as-is, I think it would also be helpful to include a similar figure showing contour plots for the taxa under iterations A-C by each investigator in the supplementary information. It would help to visualise the range of results as shown by the stress plots and maximally different contour plots here.

Still, as stated above, these suggestions are all very minor, and so I am more than happy to recommend publication following these minor revisions.

Decision letter (RSPB-2020-2809.R0)

20-Jan-2021

Dear Dr Bates

I am pleased to inform you that your manuscript RSPB-2020-2809 entitled "Evolutionary biomechanics: hard tissues and soft evidence?" has been accepted for publication in *Proceedings B*. Congratulations!!

The referee(s) have recommended publication, but also suggest some minor revisions to your manuscript. Therefore, I invite you to respond to the referee(s)' comments and revise your manuscript. Because the schedule for publication is very tight, it is a condition of publication that you submit the revised version of your manuscript within 7 days. If you do not think you will be able to meet this date please let us know. We can be flexible.

One reviewer is more critical and it would help to address their points more in the MS; but I agree with the Associate Editor that the number of investigators is sufficient.

It is a condition of publication that data supporting your paper are made available either in the electronic supplementary material or through an appropriate repository. Please see our Data Sharing Policies <https://royalsociety.org/journals/authors/author-guidelines/#data>.

[http://datadryad.org/submit?journalID=RSPB&manu=\(Document not available\)](http://datadryad.org/submit?journalID=RSPB&manu=(Document not available)) which will

take you to your unique entry in the Dryad repository. If you have already submitted your data to dryad you can make any necessary revisions to your dataset by following the above link. Please see <https://royalsociety.org/journals/ethics-policies/data-sharing-mining/> for more details.

Sincerely,

Dr John Hutchinson, Editor

Associate Editor

Board Member: 1

Comments to Author:

Thank you for submitting your manuscript to Proceedings B. Two reviewers are very complementary and have either no further comments or require some very minor corrections. A third reviewer is more critical. I believe the addition of further investigators/soft tissue reconstructions is likely unnecessary. The remainder of their comments should be addressed with some additional literature and further details in the methods.

Reviewer(s)' Comments to Author:

Referee: 1

Comments to the Author(s)

Broyde and colleagues examined the effects of interobserver variability and muscle reconstruction method on the outcomes of comparative MDA and FEA of three rodent taxa. They found substantial variation arising from observer- and reconstruction method-specific outcomes. The results clearly demonstrate a hitherto unstudied and unquantified source of error in those simulation-based methods. The paper is well written, and I have no comments to add to the paper. I applaud the authors for completing an important study to further improve the applicability and replicability of MDA and FEA as applied in functional morphological and paleontological studies.

Referee: 2

Comments to the Author(s)

Broyde and co-authors present a new methodological approach to discuss the implications about soft tissue reconstruction. It's a good idea and well-written. They use 3 common mammals as case study to evaluate problems to reconstruct soft tissues and the implications to reconstruct these tissues in fossils.

The paper seems to be the consequence of a problem arised in a previous work (Bates and Falkingham, 2018 J Anat). I agree with them that soft tissue reconstruction needs further works to obtain proper and refined results, but the submitted approach requires additional work and to be re-thought prior to be published.

On this sense, I think that the manuscript present some flaw points and possibly to re-focus the paper if authors wants to publish it on a high impact journal as Proc Roy Soc B:

-Number of investigators: this is one of the most flaw points: all work is based only on 3 investigators who independently reconstructed soft tissues. Authors recognize that this is a major limitation of the work. The bias is so important and the results can not be considered as valid just using three diferent investigators with different experience on modelling and myology.

-Iterations: it's unclear the steps followed for the 3 iterations. I recommend to add further data / figure / protocol sheet. Not sure if it's mostly based on muscle sculpture (and due to the low number of investigators previously mentioned, again, biased).

-Discussion and Conclusions: Regarding FEA implications, I'm not sure to fully agree with authors, as to perform FE analyses, it is usually required insertion areas, not using muscle volume reconstruction. This is particularly remarkable in 2D models but also in other cases, so authors need to re-evaluate the citations used to then discuss properly what it has been done in the literature.

- Discussion and Conclusions: authors include some % data about the published papers that used different variables (i.e. sensitivity) but they don't provide information about how they calculated these percentages. Is it based only on the cited publications? If yes, how can you validate that these citations are representative of all published works?

Other issues:

Line 43: "in dinosaurs and birds": As this work is focused in mammals, I think that just few citations are enough.

Line 85: "not been extensively tested", please give details (i.e. citations) about tests performed to date. Might I also recommend to down the tone of the sentence.

Referee: 3

Comments to the Author(s)

I think that this is an important and long-overdue paper. Certainly, it makes for essential - if uncomfortable - reading for those performing biomechanical modelling on extinct (or even poorly-known extant) taxa. That said, it is actually surprising how well the FEA analyses do at replicating relative stress patterns given the great uncertainty in muscle forces - this is also a very significant result (and something of a relief) in itself. The inclusion of suggestions for best practices going forwards, as drawn from quantitative results, is appreciated and maximise the utility of this paper for the field.

Fortunately, the paper is well-executed - I appreciate the author's efforts both not to over-complicate the experimental setup, and to include a varied but realistic range of investigator experience levels and data availability to replicate palaeontological analyses. I am hence happy to recommend it for publication, pending three very mild points, as listed below.

First: how accurately were lines of action estimated by each of the investigators? This is another potential source of substantial error in modelling loading regimes of extinct taxa and so, if it varied between investigators, it would be good to see it reported. I know there is less margin of error for this in rodents than in larger and taller skulls, and the consistency of fibre length results further suggest that your methodology minimised the chances of these errors, but it would still be good to verify this. Even if little-to-no variance was seen here for these reasons, it would still be worth noting that this will remain yet another source of significant source of error in other taxa (e.g. large sauropsids) beyond that seen in your small, well-bracketed mammals here.

Second, it is stated in the discussion that 0% of cranial fea studies have also applied their reconstruction techniques to extant taxa, and implied throughout that comprehensive looks at muscle properties have not been previously considered. This seems a little harsh given that, for example, Cost et al. (2020) recently combined validation work on extant lizards and parrots to ground estimates of muscle pennation angles and fibre length in their work on *Tyrannosaurus rex*. I know this was not to the same extent or with the same experimental setup or goals as you have performed here, but it at least is fair to note that some FEA work on the skulls of extinct taxa has attempted to bracket soft tissue parameters in a systematic way. (Reference: Cost, I.N. et al. (2020) Palatal biomechanics and its significance for cranial kinesis in *Tyrannosaurus rex*. *The Anatomical Record*, 303, 999-1017.

Finally, although the main text Figure 5 is currently fine as-is, I think it would also be helpful to include a similar figure showing contour plots for the taxa under iterations A-C by each investigator in the supplementary information. It would help to visualise the range of results as shown by the stress plots and maximally different contour plots here.

Still, as stated above, these suggestions are all very minor, and so I am more than happy to recommend publication following these minor revisions.

Author's Response to Decision Letter for (RSPB-2020-2809.R0)

See Appendix A.

Decision letter (RSPB-2020-2809.R1)

20-Jan-2021

Dear Dr Bates

I am pleased to inform you that your manuscript entitled "Evolutionary biomechanics: hard tissues and soft evidence?" has been accepted for publication in Proceedings B.

Open Access

Paper charges

All supplementary materials accompanying an accepted article will be treated as in their final form. They will be published alongside the paper on the journal website and posted on the online

figshare repository. Files on figshare will be made available approximately one week before the accompanying article so that the supplementary material can be attributed a unique DOI.

Sincerely,
Proceedings B
[mailto: proceedingsb@royalsociety.org](mailto:proceedingsb@royalsociety.org)

Appendix A

Author response to reviews for RSPB-2020-2809.R1: “Evolutionary biomechanics: hard tissues and soft evidence?”

We are extremely grateful to the editors and reviewers for taking the time to read and critique our manuscript so thoroughly, particularly during this challenging time. We are pleased that all are positive about what we have attempted to do in this paper, and that we have carried out the work exhaustively. We have responded to the editor and reviewer comments below. We have made modifications to our manuscript and supplementary material in response to the majority of comments but have rebutted a number of comments by reviewer 2, which we do not feel are applicable or beneficial to the study. Thank you again for taking the time to review our work.

Below, reviewer comments are in black; author responses are in blue; and quotations from the manuscript publications are in green.

Associate Editor

Board Member: 1

Comments to Author:

Thank you for submitting your manuscript to Proceedings B. Two reviewers are very complementary and have either no further comments or require some very minor corrections. A third reviewer is more critical. I believe the addition of further investigators/soft tissue reconstructions is likely unnecessary. The remainder of their comments should be addressed with some additional literature and further details in the methods.

Thank you to the editor for clearly considering both our study and the merits of specific reviewer comments, and subsequently explaining the nature of the changes that are required in our resubmission. We agree that additional modelling is unnecessary because (1) this is the first study to ever consider inter-investigator bias quantitatively (and therefore a 300% increase already on any previous study) and (2) increasing our N is unlikely to change the central message of the paper.

Reviewer(s)' Comments to Author:

Referee: 1

Comments to the Author(s)

Broyde and colleagues examined the effects of interobserver variability and muscle reconstruction method on the outcomes of comparative MDA and FEA of three rodent taxa. They found substantial variation arising from observer- and reconstruction method-specific outcomes. The results clearly demonstrate a hitherto unstudied and unquantified source of error in those simulation-based methods. The paper is well written, and I have no comments to add to the paper. I applaud the authors for completing an important study to further improve the applicability and replicability of MDA and FEA as applied in functional morphological and paleontological studies.

Thank you to reviewer 1 for taking the time to critically review our study and for their positive comments.

Referee: 2

Comments to the Author(s)

Broyde and co-authors present a new methodological approach to discuss the implications

about soft tissue reconstruction. It's a good idea and well-written. They use 3 common mammals as case study to evaluate problems to reconstruct soft tissues and the implications to reconstruct these tissues in fossils. The paper seems to be the consequence of a problem arised in a previous work (Bates and Falkingham, 2018 J Anat). I agree with them that soft tissue reconstruction needs further works to obtain proper and refined results, but the submitted approach requires additional work and to be re-thought prior to be published. On this sense, I think that the manuscript present some flaw points and possibly to re-focus the paper if authors wants to publish it on a high impact journal as Proc Roy Soc B:

We thank the reviewer for reading our study and for recognising that it addresses a fundamentally important and under-studied topic.

-Number of investigators: this is one of the most flaw points: all work is based only on 3 investigators who independently reconstructed soft tissues. Authors recognize that this is a major limitation of the work. The bias is so important and the results can not be considered as valid just using three diferent investigators with different experience on modelling and myology.

As recommended by the editor, we have not carried out any additional modelling with additional investigators. As stated above, we agree that additional modelling is unnecessary because (1) this is the first study to ever consider inter-investigator bias quantitatively (and therefore a 300% increase already on any previous study) and (2) increasing our N in unlikely to change the central message of the paper.

-Iterations: it's unclear the steps followed for the 3 iterations. I recommend to add further data / figure / protocol sheet. Not sure if it's mostly based on muscle sculpture (and due to the low number of investigators previously mentioned, again, biased).

We are slightly confused by this comment as we feel that our method is described in detail in the methods text (with reference to previous studies that have used similar methods), depicted visually in Fig 1, and then explained again in the protocol sheet in the electronic supplementary information. In our main text we state:

“Here we developed a protocol for muscle volume sculpture (Fig. 1) based on methods used in previous fossil studies [e.g. 35-36, 57, 68]. This protocol was formalised in an instruction sheet (see ESM1), which outlined the specific modelling approach to be used and anatomical diagrams on which to base the 3D muscle sculptures around 3D bone models, which are similar to those used in qualitative muscle reconstructions of fossils. As noted above, previous application of similar methods to the same fossil specimens by independent research teams have produced highly disparate muscle volumes (see discussion [37]). We therefore conducted the first analysis of inter-investigator variability in muscle volume sculpture, with three of the authors independently generating muscle volumes in all three rodent models following only the instruction sheet (ESM1).”

It should be clear from this text alone (even without looking at figure 1 and the ESM) that 3 investigators have each sculpted muscles in the 3 rodent species (3 investigators x 3 rodents = 9 iterations). We then go on to explain how these 9 muscle volume iterations were each subjected to three fibre architecture iterations, resulting in 27 fossil model iterations in total (9 volume sculptures x 3 fibre iterations = 27 fossil iterations):

However, in the absence of such data, we utilised several approaches used in a recent study [37], which cover different scenarios or assumptions about the nature of muscle architecture in the extinct group under analysis. First, we generated FLs for each muscle under the assumption that all muscles were non-pennate (i.e. parallel fibred), and that FLs were equal to muscle length (measured as the distance between the centroids of the origins and insertions in the 3D models derived from diceCT scans [70]) In this scenario, the PCSAs of all muscles are calculated according to Eq. 2 (see above). For each investigator, these models are referred to as iteration A. Second, we generated an iteration of models which differed only in their specification of the medial pterygoid muscle. This muscle consistently shows a pennate architecture in rodents [70] and in the three taxa studied here average measured pennation angles range from 20-25 degrees (Tables S1-3). Our second iteration of the models therefore represented the medial pterygoid muscle with a pennation angle of 25 degrees in all three taxa with calculated PCSA for this muscle according to Eq. 3. The average ratio of measured FL to muscle length across the three taxa was used to calculate the FLs for the medial pterygoids in this iteration (hereafter referred to as iteration B). Finally, we generated a third iteration of possible FLs and PCSAs, which are considered to be maximal reasonable deviations from the first iteration (iteration A). In this third iteration, all muscles were modelled as pennate, with a pennation angle of 25 degrees, the maximum value measured in these three rodents. The average ratio of measured FL to muscle length in each muscle across the three taxa was used to calculate the FLs for all muscles and subsequently PCSA (using Eq. 3) for this iteration (hereafter referred to as iteration C). While this might be considered an extreme deviation for the known muscle architecture of the three rodents under study, we argue this approach is important for three reasons. First, it must be acknowledged that in fossil taxa the precise values for architectural parameters are completely unknown and therefore assuming a high degree of uncertainty is the most objective approach. Second, in at least some cases, the extinct taxa under study have no direct functional analogue among extant taxa and thus their quantitative soft tissue properties may be expected to differ also. Third, at present there is relatively little quantitative data of cranial muscle architecture in extant taxa [37] and so the full range of values for extant groups are unlikely to be well sampled. These three FL and PCSA iterations were applied to the three muscle volume sculptures generated independently by the three investigators, yielding nine fossil models per taxon (27 fossil model iterations in total) to be evaluated relative to the model using real (measured) muscle values in multi-body dynamics (MDA) and finite element (FE) models.”

The other reviewers do not highlight any issues with clarity and understanding the above, and honestly, we are not sure what else we can add that would not be repetition of the information and diagrams already provided.

-Discussion and Conclusions: Regarding FEA implications, I'm not sure to fully agree with authors, as to perform FE analyses, it is usually required insertion areas, not using muscle volume reconstruction. This is particularly remarkable in 2D models but also in other cases, so authors needs to re-evaluate the citations used to then discuss properly what it has been done in the literature.

This comment isn't completely clear to us. In particular the use of the word “required”, which implies that our FE hasn't been carried out correctly or most appropriately somehow, which, to be clear, is not true. We believe the reviewer is referring to the dry skull method of deriving muscle cross sectional areas and subsequently muscle forces to apply to FEA models. This approach has been used extensively and, contrary to what the reviewer implies here, we have included many studies in our citation list and general discussion that use this

method in 2D and 3D FEA studies of fossils. Here we have focused on muscle volume sculpture approaches because (1) in the last 8-10 years there has been a big shift towards this approach and away from the dry skull methods in macroevolutionary studies (particularly in “high impact” studies of major evolutionary transitions), and (2) muscle volume sculpture has been employed across the board in studies of different aspects of anatomy (limbs, axial and skulls) and modelling techniques (MDA, FEA and by itself to quantify muscle size evolution). By contrast, the dry skull method is used exclusively on skulls and nearly always only for FEA studies. Therefore, we used muscle sculpture because it has both the broadest and most significant impact/importance. To our knowledge the accuracy of the dry skull method has only been objectively tested in one species in one study (Law & Rita 2019, *J. Morphology* 280: 1706-1713) where it was shown to be fairly inaccurate. While we agree that it would be interesting to assess the dry skull method using our models, we feel quite strongly that it is beyond the scope of the current study which already contains a very large number of fossil iterations and a lot of new data. We have, however, modified one sentence in the introduction to more explicitly referred to the dry skull method and added citations to each approach to refer the reader to relevant literature more directly (additions in red):

“Biomechanical modelling studies of extinct animals have subsequently employed a diverse range of approaches to estimate absolute values for soft tissue parameters in fossil organisms, ranging from standardised properties based on estimated mean values for living taxa [e.g. 12-13,19-21,34], scaling values from analogous extant animals [e.g. 12-13, 24-26, 28-31], extrapolating values from estimated muscle attachment areas [e.g. 10,27,37-38] and computer-aided design approaches to reconstruct the size and geometry of soft tissues directly in the fossil themselves [e.g. 5, 14-16, 22-23, 34-36].”

- Discussion and Conclusions: authors include some % data about the published papers that used different variables (i.e. sensitivity) but they don't provide information about how they calculated these percentages. Is it based only on the cited publications? If yes, how can you validate that these citations are representative of all published works?

Contrary to this comment, in our first submission we explicitly stated that our percentages were based on a subjective evaluation of specific studies that we cited, and we described the basis on which we including those papers:

“To put our study and its conclusions into context, we surveyed 67 published studies that utilised quantitative soft tissue reconstruction alone or in combination with biomechanical models to examine evolutionary changes in functional morphology in fossil taxa [2-32, 35-69]. Our goal was not to provide exhaustive coverage of all relevant papers, but to sample enough studies to provide coverage of most major taxonomic groups, body regions (limbs, skulls, necks etc.) and methodological approaches.”

While we state here that our goal was not to provide exhaustive coverage of all published work, we did actually make a genuine effort to find every paper we could where the results hinged on soft tissue reconstruction in a fossil. This meant we didn't include, for example, every single FEA fossil skull study because not all these studies took an analytical approach that required soft tissue reconstruction. For example, some studies scaled all skulls/jaw to the same size and then applied a uniform hypothetical load (e.g. 10 Newtons) to all of them and compared the results. These kinds of studies are not relevant here and were not included given the criteria we state in the relevant paragraph in the discussion. Ultimately we acknowledge that this is our subjective assessment of current literature that exists now (which

we cite) and some readers may disagree, and any assessment will be subject to future change (as demonstrated by the paper that reviewer 3 cites on this point below, which came out after we submitted our manuscript).

Other issues:

Line 43:"in dinosaurs and birds": As this work is focused in mammals, I think that just few citations are enough.

We need to cite these studies anyway for our discussion and evaluation of the wider literature in the Discussion & Conclusion section so we don't seem any harm in acknowledging this work here too. It would seem odd to cite studies within the paper as a whole, but not to cite some them in specific contexts that they are relevant.

Line 85: "not been extensively tested", please give details (i.e. citations) about tests performed to date. Might I also recommend to down the tone of the sentence.

We think the reviewer means line 95. Again, we disagree with this comment. We have previously discussed how palaeontologists have sought to address uncertainty in soft tissue reconstruction at length in the introduction prior to this concluding sentence. For example:

“Biomechanical modelling studies of extinct animals have subsequently employed a diverse range of approaches to estimate absolute values for soft tissue parameters in fossil organisms, ranging from standardised properties based on estimated mean values for all living taxa, scaling values from supposed analogous extant animals, and computer-aided design approaches to reconstruct the size and geometry of soft tissues directly in the fossil themselves. Sensitivity analyses have been carried out in small number of these studies and have consistently shown that large errors in soft tissue parameters will lead to significant inaccuracy in function or performance predictions [12-14, 20, 22, 36-37]. However, it remains qualitatively and quantitatively uncertain what the likely error magnitudes are for such soft tissue reconstructions.”

These sentences already achieve what the reviewer is asking for above, including citing relevant studies. We are happy that the sentence we presume the reviewer is referring to on line 95 is an appropriate bridge between the introduction and methods section.

Referee: 3

Comments to the Author(s)

I think that this is an important and long-overdue paper. Certainly, it makes for essential - if uncomfortable - reading for those performing biomechanical modelling on extinct (or even poorly-known extant) taxa. That said, it is actually surprising how well the FEA analyses do at replicating relative stress patterns given the great uncertainty in muscle forces - this is also a very significant result (and something of a relief) in itself. The inclusion of suggestions for best practices going forwards, as drawn from quantitative results, is appreciated and maximise the utility of this paper for the field.

We thank the reviewer for carefully considering of study and recognising its timely and important nature.

Fortunately, the paper is well-executed - I appreciate the author's efforts both not to over-

complicate the experimental setup, and to include a varied but realistic range of investigator experience levels and data availability to replicate palaeontological analyses. I am hence happy to recommend it for publication, pending three very mild points, as listed below.

Thank you. There is a huge amount of data in this paper and it was challenge to present it all succinctly.

First: how accurately were lines of action estimated by each of the investigators? This is another potential source of substantial error in modelling loading regimes of extinct taxa and so, if it varied between investigators, it would be good to see it reported. I know there is less margin of error for this in rodents than in larger and taller skulls, and the consistency of fibre length results further suggest that your methodology minimised the chances of these errors, but it would still be good to verify this. Even if little-to-no variance was seen here for these reasons, it would still be worth noting that this will remain yet another source of significant source of error in other taxa (e.g. large sauropsids) beyond that seen in your small, well-bracketed mammals here.

The reviewer is absolutely correct in their assertion that muscle lines of actions have an important impact on the bite forces output by these models. This has been demonstrated specifically by co-author Fagan's previous work on extant taxa and Bates' work on dinosaur limbs. However, muscle geometry/line of action was not tested explicitly here. Rather it was standardised across the investigators and therefore didn't vary. We have tried to explain this more explicitly in the methods in the resubmission, e.g.

“Muscle geometries (origins, insertions and approximate lines of action) were based on physical dissection and contrast-enhanced micro-CT reconstructions of the specimens being modelled [70] and were standardised across all model iterations.”

So basically, because the muscle lines of action are always the same and based on high resolution data from the animals themselves, they are (1) likely to be very accurate and (2) don't vary and therefore contribute to the variation we present across the model iterations. We apologise for not being clearer in our original submission.

Second, it is stated in the discussion that 0% of cranial fea studies have also applied their reconstruction techniques to extant taxa, and implied throughout that comprehensive looks at muscle properties have not been previously considered. This seems a little harsh given that, for example, Cost et al. (2020) recently combined validation work on extant lizards and parrots to ground estimates of muscle pennation angles and fibre length in their work on Tyrannosaurus rex. I know this was not to the same extent or with the same experimental setup or goals as you have performed here, but it at least is fair to note that some FEA work on the skulls of extinct taxa has attempted to bracket soft tissue parameters in a systematic way. (Reference: Cost, I.N. et al. (2020) Palatal biomechanics and its significance for cranial kinesis in Tyrannosaurus rex. The Anatomical Record, 303, 999-1017.

We thank the reviewer for bringing this excellent recent study to our attention. We agree that it certainly should be referenced here, and we have made appropriate modifications to the discussion and percentages quoted.

Finally, although the main text Figure 5 is currently fine as-is, I think it would also be helpful to include a similar figure showing contour plots for the taxa under iterations A-C by each

investigator in the supplementary information. It would help to visualise the range of results as shown by the stress plots and maximally different contour plots here.

We thank the reviewer for this suggestion and agree that it will be a useful addition to the supplementary information. We have therefore added these images (Fig S5 in the ESM).

Still, as stated above, these suggestions are all very minor, and so I am more than happy to recommend publication following these minor revisions.